# Effects of Electrocardiographic Monitoring Education on Nurses’ Confidence and Psychological Stress: An Online Cross-Sectional Survey in Japan

**DOI:** 10.3390/ijerph19084742

**Published:** 2022-04-14

**Authors:** Sho Nishiguchi, Nagisa Sugaya, Yusuke Saigusa, Michinori Mayama, Takuhiro Moromizato, Masahiko Inamori, Yasuharu Tokuda, Takashi Watari

**Affiliations:** 1Department of General Internal Medicine, Shonan Kamakura General Hospital, Kamakura 247-8533, Japan; 2Unit of Public Health and Preventive Medicine, Yokohama City University, Yokohama 236-0027, Japan; nagisa_s@yokohama-cu.ac.jp; 3Department of Internal Medicine, Hayama Heart Center, Miura 240-0116, Japan; 4Department of Biostatistics, School of Medicine, Yokohama City University, Yokohama 236-0027, Japan; saigusay@yokohama-cu.ac.jp; 5Department of Obstetrics and Gynecology, Graduate School of Medicine, Hokkaido University, Sapporo 060-0814, Japan; marimo.mayama@gmail.com; 6Renal & Rheumatology Division, Internal Medicine Department, Okinawa Prefectural Nambu Medical Center & Children Medical Center, Okinawa 901-1193, Japan; tmoromizato@gmail.com; 7Department of Medical Education, Yokohama City University, Yokohama 236-0027, Japan; inamorim@yokohama-cu.ac.jp; 8Muribushi Okinawa Center for Teaching Hospitals, Urasoe 901-2132, Japan; yasuharu.tokuda@gmail.com; 9General Medicine Center, Shimane University Hospital, Izumo 693-8501, Japan; wataritari@gmail.com; 10Devision of Hospital Medicine, University of Michigan Health System, Ann Arbor, MI 48105, USA

**Keywords:** nurse, education, continuous electrocardiography monitoring, confidence, psychological stress

## Abstract

We aimed to investigate the association between nurses’ electrocardiographic (ECG) monitoring education and their confidence and psychological stress regarding ECG monitoring. In 2019, a web-based cross-sectional study was conducted among Japanese nurses. A multivariable logistic regression analysis was performed to evaluate the effects of education on nurses’ confidence and psychological stress regarding ECG monitoring. In total, 1652 nurses were included in the study. Factors significantly associated with nurses’ confidence were post-graduate education experience (odds ratio [OR], 2.4; 95% confidence interval [CI], 1.6–3.6), ≥11 post-graduate years (OR, 2.2; 95% CI, 1.5–3.1), male gender (OR, 4.4; 95% CI, 2.9–6.6), ≥5 helpful experiences with ECG monitoring (OR, 10.7; 95% CI, 6.0–19.1), work experience in an intensive care unit (OR, 2.3; 95% CI, 1.5–3.7), and work experience in a cardiology department (OR, 1.7; 95% CI, 1.2–2.4). Factors significantly associated with nurses’ psychological stress were male gender (OR, 1.9; 95% CI, 1.2–2.9), ≥5 helpful experiences with ECG monitoring (OR, 1.9; 95% CI, 1.2–2.9), and work experience in an emergency room (OR, 2.4; 95% CI, 1.3–4.8). These results suggest that nurses’ post-graduate ECG monitoring education enhanced their confidence, but did not reduce psychological stress regarding ECG monitoring.

## 1. Introduction

Electrocardiographic (ECG) monitoring was first introduced into medical practice over 60 years ago [1]. The simple three-electrode lead monitoring of patients is common in hospitalised patient care, such as continuous ECG monitoring. Continuous ECG monitoring is expected for patients in critical care units and is commonly used in progressive or step-down units; in Japan, these are also known as high care units (HCU). Furthermore, ECG monitoring is used for some patients in the emergency department as well as other inpatient units. Due to technological advances in ECG monitoring, healthcare professionals, including nurses, can now track a patient’s heart rate with the basic rhythm and detect arrhythmia, myocardial ischaemia, and prolonged QT syndrome. These life-threatening situations, reflected by abnormalities that can be detected by ECG monitoring, require quick diagnosis and treatment.

Nurses play an important role in the management of patients’ ECG monitoring. They perform the following difficult roles,: assessing which patients require ECG monitoring, correctly placing ECG monitoring equipment on patients, quickly detecting abnormal data, and deciding whether to intervene in clinical practice. Nurses put three-electrode lead continuous ECG monitors on their inpatients based on doctors’ orders and they monitor patients at the nurses’ station. Nurses are often the first care providers to evaluate hospitalised patients presenting with abnormalities in ECG monitoring via monitoring alarms [2]. In a previous systematic review, intensive care nurses feel alarm fatigue with psychological stress and there is no clear system for managing the alarms of the monitoring devices [3]. In a Chinese national survey, more than half of nurses indicated that they are not adequately skilled in ECG monitoring to detect myocardial ischemia [4]. They have an enormous responsibility and must make important clinical decisions related to cardiac data monitoring every day. Therefore, nurses must be repeatedly educated and trained in ECG monitoring [2].

Nurses receive different post-graduate ECG monitoring education based on each nurse’s place of employment and may have been provided with insufficient pre-graduate ECG monitoring education for application in clinical practice. Post-graduate ECG monitoring education content varies; ECG monitoring practices are inconsistent among nurses [5]. The American Heart Association proposed practice standards for improving hospital ECG monitoring in 2004 [6,7], mainly as post-graduate ECG monitoring education. Evidence shows that cardiology departments that implemented the American Heart Association practice standards for ECG monitoring training among nurses witnessed improved patient outcomes [2]. However, the settings of these studies were limited to specialized departments, and there have been few epidemiological studies on nationwide post-graduate ECG monitoring education for nurses. Japanese Ministry of Health, Labour and Welfare introduced The Bachelor of Nursing program at 255 universities in 2017 in order to increase the levels of nursing school education available [7]. Based on the program, each nursing school creates an education curriculum, but ECG monitoring is not a required course in the Bachelor of Nursing program. Therefore, the contents of pre-graduate ECG monitoring education differ among nursing schools. Nurses rarely receive pre-graduate ECG monitoring education. Furthermore, very few national surveys have examined this topic [8].

We hypothesized that without an ECG monitoring education, nurses cannot gain confidence, and they may thus find it stressful to care for patients using continuous ECG monitoring. Nursing requires the delivery of humane, empathetic, culturally sensitive, proficient, and moral care in work environments with limited resources and increasing responsibilities. Thus, among nurses, any imbalance between providing high-quality care and coping with stressful work could lead to burnout [9]. It is well known that work-related stress is related to burnout [10,11]. Thus, we hypothesise that, among nurses, effective pre- and post-graduate education is essential for gaining confidence and reducing the psychological stress related to caring for patients using continuous ECG monitoring. To the best of our knowledge, no studies have examined the effect of pre- and post-graduate education on nurses’ confidence and stress related to continuous ECG monitoring.

The present study aimed to investigate nurses’ ECG monitoring education in terms of (1) the relationship between nurses’ education and their confidence and psychological stress regarding continuous ECG monitoring, and (2) the actual situation of nurses’ pre- and post-graduate ECG monitoring education.

## 2. Materials and Methods

### 2.1. Participants and Setting

We conducted a cross-sectional study using a web-based questionnaire for nurses from March to August 2019. The study was conducted in collaboration with NURSE SENKA, a publishing company that supplies materials for nurses and nursing students in Japan. From among the 188,701 readers of the company’s publications at the time of the study, including enrolled nurses and nursing students, we recruited 2500 participants using a web-based survey for collaborative descriptive nursing research [12]. Before initiation of the joint research study, we calculated the number of participants based on seeing the numbers on the NURSE SENKA website (https://nursepress.jp/, accessed on 28 January 2022). The study’s margin of error was plus or minus 2%. Our questionnaire was sent through an email invitation and posted via an online link on the NURSE SENKA website, which provides information about nursing skills and medical topics to its readers, see Appendix A. The collaborative research examined nursing education regarding the monitoring of vital signs and the relationship between nurses’ menstruation status and working patterns. To increase participant recruitment, we announced that the participants would have an opportunity to obtain points from the publisher. Among the applicants who responded to the web-based survey, one person would receive points equivalent to 10,000 yen two people would receive points equivalent to 5000 yen (37.6 £), ten would receive points equivalent to 1000 yen and twenty would receive points equivalent to 500 yen In the present study, we excluded nursing students and nurses who did not perform ECG monitoring in practice, including nursing directors and nurses who worked in outpatient care, home medical care, and long-term care.

### 2.2. Data Collection

For the completeness of data, the questionnaire format did not allow participants to proceed to the next page if they left questions unanswered. The following data were collected from each nurse: number of post-graduate years, gender, nursing license type (state-registered or state-enrolled nurse), hospital scale, managerial position, department, pre-graduate education, post-graduate education, helpful experiences with ECG monitoring, and the contributions of pre- and post-graduate education to patient care (useful, modestly useful, minimally useful, or not useful). Pre-graduate education referred to the provision of ECG monitoring education during nursing school. Post-graduate education referred to the provision of ECG monitoring education after graduation from nursing school. The classification of helpful experiences with ECG monitoring was based on a nurse’s number of experiences in saving a patient’s life owing to continuous ECG monitoring while caring for patients. Helpful experiences with ECG monitoring were categorised as one to four times and ≥five times. The contributions of pre- and post-graduate education to patient care were classified according to four levels based on the nurses’ subjective assessment: not useful, minimally useful, modestly useful, or useful.

In the data collected above, postgraduate years were categorised as ≤5 years, 6–10 years, and ≥11 years. The nursing license type was either a state-registered or state-enrolled nurse. In Japan, the differences between state-registered and state-enrolled nursing licenses are education time (over 3000 h and 1800 h, respectively) and eligibility requirements for admission to each type of nursing school (graduation from high school and graduation from junior high school, respectively). Hospital scale was classified according to the number of hospital beds: ≤99 beds, 100–299 beds, and ≥300 beds. We identified the nurses’ positions as either managerial or non-managerial. The departments were internal medicine, surgery, paediatrics, obstetrics and gynaecology, ER, ICU/HCU, dermatology, ophthalmology, otolaryngology, urology, rehabilitation, psychiatry, gastroenterology, cardiology, nephrology, diabetes and endocrinology, rheumatology and clinical immunology, allergy, haematology, neurology, psychosomatic medicine, infectious disease medicine, oncology, thoracic surgery/cardiovascular surgery/respiratory surgery, breast surgery, thyroid surgery, paediatric surgery, proctology, orthopaedic surgery, neurosurgery, and plastic surgery.

### 2.3. Data Analysis

We recruited 2500 people for this collaborative descriptive nursing research [13]. A multivariable logistic analysis was conducted to identify the relationship between pre- and post-graduate ECG monitoring education and nurses’ confidence and psychological stress regarding continuous ECG monitoring. The primary outcomes were the presence or absence of nurses’ confidence and psychological stress regarding continuous ECG monitoring. They responded to questions on a four-point scale of feelings on the questionnaire sheet; the four-point scale was used for ease of response. The scale ranged from 1 = ‘not confident at all’ to 4 = ‘confident’ and from 1 = ‘very stressful’ to 4 = ‘not stressful at all’. Furthermore, we grouped the nurses into those who felt confident, those who did not, those who experienced psychological stress, and those who did not, in order to perform a logical analysis. For the sub-group analysis of nurses who received post-graduate ECG monitoring education, a statistical analysis was performed using multivariable analysis. In the model for analysing nurses who received post-graduate education, useful post-graduate education was included as a factor. A two-tailed *p*-value < 0.05 was considered statistically significant. All data analyses were performed using SPSS Statistics version 21 J (IBM, Tokyo, Japan).

### 2.4. Ethical Consideration

This study was approved by the Institutional Review Board of Shimane University (No. 3515 [approved on 16 January 2019]). The study was conducted in accordance with the principles of the Declaration of Helsinki. The nurses’ consent was implied in their responses to the questionnaires. Before the initiation of answering a web-based questionnaire, information on the study objective, data collection, researchers’ names and affiliations, and data anonymity were presented on the website.

## 3. Results

### 3.1. Baseline Findings

Among the 2500 consecutive participants, 287 nursing students and 561 nurses who did not perform continuous ECG monitoring in their work were excluded (five nurses were nursing directors, 304 nurses worked in outpatient care, 96 nurses worked in home visit care, and 156 nurses worked in long-term care). Therefore, this study included 1652 nurses.

### 3.2. Characteristics and Univariable Analysis

As shown in Table 1, nurses with ≥11 post-graduate years accounted for 44.3% of the sample, and almost all nurses were female (90.6%). Almost all nurses had state-registered nursing licenses (93.7%), and 82.4% of nurses worked night shifts (Table 1). More than half of the nurses (53.5%) worked in large-scale hospitals (Table 1). Most nurses (72.4%) had one to four or ≥five helpful experiences with continuous ECG monitoring in patient care (Table 1).

### 3.3. The Situation of Pre- and Post-Graduate ECG Monitoring Education

Figure 1 shows that 47.9% of the nurses studied ECG monitoring at nursing school, and 72.3% of the nurses received post-graduate ECG monitoring education. Figure 1 also indicates that only 37.0% of nurses received both pre- and post-graduate ECG monitoring education, while 52.1% of nurses reported receiving no pre-graduate ECG monitoring education. Among the participants who received pre-graduate ECG monitoring education, 57.4% responded that the education was minimally useful or not useful in contributing to patient care (Figure 2). However, among the participants who received post-graduate ECG monitoring education, 85.7% reported that the education was useful in contributing to patient care (Figure 2).

### 3.4. Effects of Pre- and Post-Graduate ECG Monitoring Education

We analysed the relationship between pre- and post-graduate ECG monitoring education and nurses’ confidence and stress regarding continuous ECG monitoring. The univariable analysis revealed that post-graduate ECG monitoring education was significantly associated with nurses’ confidence regarding continuous ECG monitoring (Table 1). As shown in Table 2, the multivariable analysis showed that pre-graduate ECG monitoring education was not associated with nurses’ confidence regarding continuous ECG monitoring, but post-graduate ECG monitoring education was significantly associated with nurses’ confidence in continuous ECG monitoring (adjusted odds ratio [OR], 2.40; 95% confidence interval [CI], 1.59–3.61; *p* < 0.001). Table 3 shows that pre- and post-graduate ECG monitoring education was not significantly associated with nurses’ psychological stress regarding continuous ECG monitoring.

### 3.5. Effects of Useful Post-Graduate Education on Nurses’ Confidence Regarding ECG Monitoring

Among the nurses who received post-graduate ECG monitoring education, we focused on the association between the significant factor of post-graduate ECG monitoring education and nurses’ confidence regarding continuous ECG monitoring. As shown in Table 4, the useful contributions of post-graduate education toward improving patient care were significantly associated with nurses’ confidence regarding continuous ECG monitoring (adjusted OR, 3.96; 95% CI, 2.85–5.51; *p* < 0.001).

### 3.6. Other Factors Associated with Nurses’ Confidence and Psychological Stress Regarding Continuous ECG Monitoring

Multivariable analysis of nurses’ confidence regarding continuous ECG monitoring showed that, in addition to post-graduate ECG monitoring education, male gender (adjusted OR, 4.38; 95% CI, 2.90–6.62; *p* < 0.001), ≥11 post-graduate years (adjusted OR, 2.15; 95% CI, 1.49–3.10; *p* < 0.001), helpful experience with continuous ECG monitoring (one to four times: adjusted OR, 4.07; 95% CI, 2.30–7.22; *p* < 0.001; ≥five times: adjusted OR, 10.69; 95% CI, 5.98–19.13; *p* < 0.001), work experience in the ICU or HCU (adjusted OR, 2.31; 95% CI, 1.45–3.68; *p* < 0.001), and work experience in the cardiology department (adjusted OR, 1.70; 95% CI, 1.19–2.43; *p* = 0.004) were significantly associated factors (Table 2). Moreover, female gender (adjusted OR, 0.53; 95% CI, 0.34–0.82; *p* = 0.004), ≥5 helpful experiences with continuous ECG monitoring (adjusted OR, 0.53; 95% CI, 0.34–0.84; *p* = 0.007), and work experience in the ER department (adjusted OR, 0.41; 95% CI, 0.34–0.82; *p* = 0.004) were significantly associated with a reduction in nurses’ psychological stress regarding continuous ECG monitoring (Table 3).

## 4. Discussion

The present study revealed the following two findings: (1) less than half of the nurses received pre-graduate ECG monitoring education (47.9%), while 72.3% of the nurses received post-graduate ECG monitoring education as part of the whole department (Figure 1); and (2) the experience of post-graduate ECG monitoring education was significantly related to nurses’ confidence regarding continuous ECG monitoring, while experience of pre- and post-graduate education had no effect on nurses’ psychological stress regarding continuous ECG monitoring in the multivariable analyses.

The present study’s findings revealed that Japanese nursing schools have not offered adequate opportunities for pre-graduate ECG monitoring education. The previous literature does not include any large-scale descriptive studies on pre-graduate ECG monitoring education. Regarding the reasons for the availability of relatively few pre-graduate education ECG monitoring education opportunities, we speculate that, due to insufficient education time during nursing school and reporting bias, there may be fewer reports of such opportunities compared to the actual number of existing opportunities. Nursing schools should thus increase such opportunities in pre-graduate ECG monitoring education.

Among the nurses who received pre-graduate ECG monitoring education, more than half (57.4%) responded that such education was not useful or was minimally useful in contributing to patient care (Figure 2). The study results showed that teaching practical and critical content using lectures in pre-graduate education settings can be challenging. Thus, teachers must plan how to impart ECG monitoring education to nursing students [14].

This study showed that a large number of nurses received post-graduate ECG monitoring education (72.3%). While our study included nurses from various departments, this finding is similar to that of a Chinese study (74.3%) that was limited to a cardiology department [4]. Previous research regarding ECG monitoring education in the nursing education field has been conducted in most settings, including the cardiology department, surgical department, stroke unit, ER, and ICU [4,15,16,17,18,19,20,21]. In this study, among the nurses who received post-graduate ECG monitoring education, a larger percentage indicated that such education was modestly useful (47.9%) compared to those who indicated that it was useful (37.8%) (Figure 2). Thus, post-graduate ECG monitoring education is essential to improve the quality of educational content in many hospitals.

The present research results indicate that post-graduate ECG monitoring education can affect nurses’ confidence regarding continuous ECG monitoring. In a study, high-fidelity simulation education related to confidence regarding nursing skills, but these skills didn’t include ECG monitoring [22]. However, to the best of our knowledge, no previous research has examined the effect of post-graduate ECG monitoring education on nurses’ confidence regarding continuous ECG monitoring. The sub-group analysis showed that, among those nurses who received post-graduate ECG monitoring education, useful post-graduate education was related to nurses’ confidence regarding continuous ECG monitoring. Useful post-graduate ECG monitoring education could be achieved by implementing ECG monitoring in standard practice; these include correct electrode placement, consideration of the appropriateness of the order for monitoring, and the identification of critical cardiac abnormalities [4,19,23]. On the other hand, the study results showed that post-graduate ECG monitoring education was not directly related to nurses’ psychological stress regarding continuous ECG monitoring. These results indicate that effective post-graduate ECG monitoring for nurses may contribute to their greater confidence regarding continuous ECG monitoring. Future prospective studies are expected.

Only the male gender and helpful experiences with continuous ECG monitoring were associated with both nurses’ confidence and reduced psychological stress regarding continuous ECG monitoring (Table 2 and Table 3). In general, there are fewer male than female nurses in the profession, and male nurses tend to be employed in ‘fast-paced, high-tech areas’ such as the ICU and ER [24,25,26]. However, no previous studies have revealed the relationship between gender differences and nurses’ confidence regarding continuous ECG monitoring. It is reasonable that helpful experiences with continuous ECG monitoring were associated with nurses’ confidence and reduced psychological stress. We suggest that post-graduate ECG monitoring education could contribute to increasing the number of helpful experiences with continuous ECG monitoring; consequently, as a result of these experiences, nurses gain confidence and experience reduced psychological stress regarding continuous ECG monitoring. We hope that this hypothesis will be proven in future research.

This study also identified other factors associated with nurses’ confidence and psychological stress regarding ECG monitoring. It is unsurprising that nurses with ≥11 post-graduate years had more confidence than nurses with fewer postgraduate years. Nurses who have work experience in the ICU/HCU or cardiology departments may tend to be interested in continuous ECG monitoring because of these specialties. ER nurses tend to experience less stress than nurses in other departments because they tend to work with critical patients, including those with ischaemic heart disease and arrhythmia.

This web-based research recruited nurses who worked all over Japan. The present study, as a Japanese descriptive study, was the first to report the insufficient provision of pre-graduate ECG monitoring education, and showed that the key challenge of post-graduate ECG monitoring education may improve its quality so that it is useful in nursing practice. Moreover, our research’s new findings are valuable for showing that post-graduate ECG monitoring education, especially useful post-graduate education, could contribute toward improving nurses’ confidence regarding continuous ECG monitoring. Based on the present study findings, prospective interventional research on post-graduate ECG monitoring education and its relationship with nurses’ confidence should be conducted in the near future.

This study has several limitations. First, it does not establish causal relationships due to its retrospective study design. Second, a web-based survey was performed without participant supervision; however, the research study may still have a social desirability bias [27]. Third, the main strength of the web-based survey was its data completeness; however, the response rate in the population of unselected participants was low [28]. Fourth, among our research participants, we were unable to differentiate those attending three- or four-year schools for state-registered nursing licenses. Fifth, the study data were collected through a web-based survey that included rewards based on participation, and random sampling could not be performed; thus, the representativeness of the sample could not be guaranteed and recruitment bias may be a factor due to the rewards. The sample we collected could not be matched to the proportions of each age group in each region throughout Japan.

## 5. Conclusions

Post-graduate ECG monitoring education experience was associated with nurses’ improved confidence regarding continuous ECG monitoring. However, the experience of post-graduate education had no effect on nurses’ psychological stress regarding continuous ECG monitoring. Less than half of the study participants received pre-graduate ECG monitoring education. Pre-graduate education can provide more chances for nurses to learn ECG monitoring, and post-graduate education may contribute to their confidence.

## Figures and Tables

**Figure 1 ijerph-19-04742-f001:**
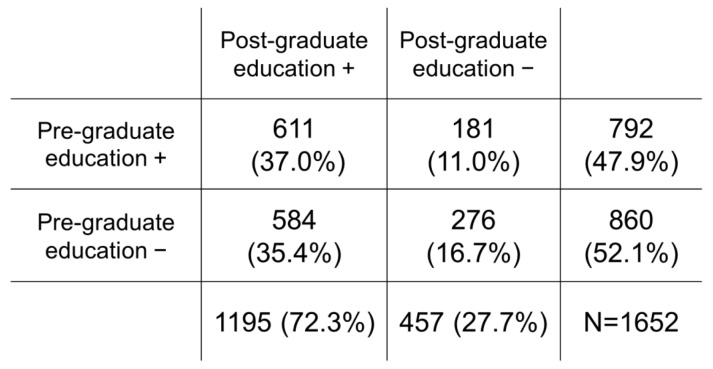
Nurses’ pre- and post-graduate ECG monitoring education (the numbers and the percentages of all participants).

**Figure 2 ijerph-19-04742-f002:**
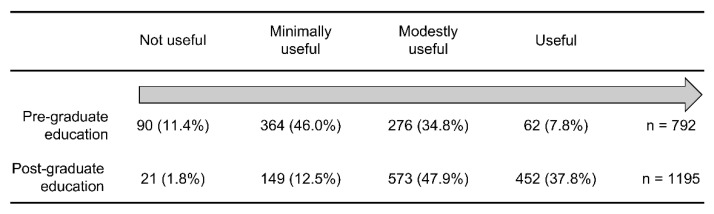
Contributions of pre- and post-graduate education to patient care (the numbers and percentages of total participants in each education group).

**Table 1 ijerph-19-04742-t001:** Univariable analysis of nurses’ confidence and psychological stress regarding continuous ECG monitoring.

N = 1652	n (%)	Confidence to ECG Monitoring	Stress to ECG Monitoring
Pre-graduate education	792 (47.9)	*p* = 0.269	*p* = 0.475
Post-gradute education	1195 (72.3)	*p* < 0.001 *	*p* = 0.280
State registered nurse {vs. State enrolled nurse}	1548 (93.7)	*p* < 0.001 *	*p* = 0.324
Male gender nurse	156 (9.4)	*p* < 0.001 *	*p* < 0.001 *
Postgraduted year [1–5, 6–10, ≥11 years]	599, 320, 732 (36.3, 19.4, 44.3)	*p* < 0.001 *	*p* = 0.407
Night shift	1362 (82.4)	*p* < 0.001 *	*p* = 0.607
Helpful experience from ECG monitoring [0, 1–4, ≥5 times]	456, 763, 433 (27.6, 46.2, 26.2)	*p* < 0.001 *	*p* < 0.001 *
Hospital scale [≤99, 100–299, ≥300 beds]	141, 627, 884 (8.5, 38.0, 53.5)	*p* = 0.008 *	*p* = 0.631
No manager position	898 (54.4)	*p* = 0.397	*p* = 0.902
Critical care nursing	4 (0.2)	*p* = 0.254	*p* = 0.889
ICU, HCU	117 (7.1)	*p* < 0.001 *	*p* < 0.001 *
ER	58 (3.5)	*p* < 0.001 *	*p* = 0.003 *
Cardiology department	383 (23.2)	*p* < 0.001 *	*p* = 0.763
Thoracic surgery, Cardiovascular surgery, Respiratory surgery department	185 (11.2)	*p* < 0.001 *	*p* = 0.389

ECG: electrocardiogram; ICU: intensive care unit; HCU: high care unit; ER: emergency room; Based on Chi-square analysis, * *p* < 0.05.

**Table 2 ijerph-19-04742-t002:** Multivariable analysis of nurses’ confidence regarding continuous ECG monitoring.

N = 1652	Odds	CI	*p*-Value
Pre-graduate education	0.96	(0.71–1.28)	*p* = 0.764
Post-gradute education	2.40	(1.59–3.61)	*p* < 0.001 **
State registered nurse	2.73	(1.05–7.14)	*p* = 0.040 *
Male gender nurse	4.38	(2.90–6.62)	*p* < 0.001 **
Postgraduated year			*p* < 0.001 **
6–10 years	1.52	(0.98–2.36)	*p* = 0.060
≥11 years	2.15	(1.49–3.10)	*p* < 0.001 **
Night shift	1.30	(0.86–1.99)	*p* = 0.217
Helpful experience from ECG monitoring			*p* < 0.001 **
1–4 times	4.07	(2.30–7.22)	*p* < 0.001 **
≥5 times	10.69	(5.98–19.13)	*p* < 0.001 **
No manager position	1.29	(0.96–1.72)	*p* = 0.090
Hospital scale			*p* = 0.099
100–299 beds	1.34	(0.74–2.44)	*p* = 0.340
≥300 beds	1.73	(0.96–3.13)	*p* = 0.069
ICU, HCU	2.31	(1.45–3.68)	*p* < 0.001 **
ER	2.26	(1.18–4.32)	*p* = 0.014 *
Cardiology department	1.70	(1.19–2.43)	*p* = 0.004 **
Thoracic surgery, Cardiovascular surgery, Respiratory surgery department	1.76	(1.13–2.74)	*p* = 0.012 *

ECG: electrocardiogram; CI: confidence interval; ICU: intensive care unit; HCU: high care unit; ER: emergency room. Based on logistic analysis, *: *p* < 0.05; **: *p* < 0.01.

**Table 3 ijerph-19-04742-t003:** Multivariable analysis of nurses’ psychological stress regarding continuous ECG monitoring.

N = 1652	Odds	CI	*p*-Value
Pre-graduate education	0.97	(0.71–1.33)	*p* = 0.840
Post-gradute education	0.98	(0.68–1.42)	*p* = 0.926
State registered nurse	0.61	(0.27–1.36)	*p* = 0.228
Male gender nurse	0.53	(0.34–0.82)	*p* = 0.004 **
Postgraduated year			*p* = 0.021 *
6–10 years	1.67	(1.06–2.63)	*p* = 0.028 *
≥11 years	1.58	(1.10–2.29)	*p* = 0.014 *
Night shift	1.20	(0.79–1.82)	*p* = 0.402
Helpful experience from ECG monitoring			*p* = 0.003 *
1–4 times	0.98	(0.65–1.48)	*p* = 0.907
≥5 times	0.53	(0.34–0.84)	*p* = 0.007 **
No manager position	1.09	(0.80–1.49)	*p* = 0.569
Hospital scale			*p* = 0.615
100–299 beds	0.80	(0.43–1.51)	*p* = 0.497
≥300 beds	0.94	(0.50–1.76)	*p* = 0.838
ICU, HCU	0.52	(0.31–0.87)	*p* = 0.012 *
ER	0.41	(0.21–0.79)	*p* = 0.008 **
Cardiology department	1.04	(0.69–1.56)	*p* = 0.859
Thoracic surgery, Cardiovascular surgery, Respiratory surgery department	0.76	(0.46–1.26)	*p* = 0.285

ECG: electrocardiogram; CI: confidence interval; ICU: intensive care unit; HCU: high care unit; ER: emergency room. Based on logistic analysis, * *p* < 0.05; ** *p* < 0.01.

**Table 4 ijerph-19-04742-t004:** Sub-analysis of factors associated with nurses’ confidence regarding continuous ECG monitoring among nurses who received post-graduate ECG monitoring education.

n = 1194	Odds	CI	*p*-Value
Pre-graduate education	0.79	(0.57–1.09)	*p* = 0.151
Useful post-gradute education	3.96	(2.85–5.51)	*p* < 0.001 **
State registered nurse	2.30	(0.75–7.05)	*p* = 0.144
Male gender nurse	3.99	(2.48–6.41)	*p* < 0.001 **
Postgraduated year			*p* = 0.001 **
6–10 years	1.68	(1.03–2.75)	*p* = 0.040 *
≥11 years	2.23	(1.46–3.39)	*p* < 0.001 **
Night shift	1.15	(0.72–1.83)	*p* = 0.550
Helpful experience from ECG monitoring			*p* < 0.001 **
1–4 times	4.04	(2.00–8.19)	*p* < 0.001 **
≥5 times	9.63	(4.73–19.61)	*p* < 0.001 **
No manager position	1.10	(0.79–1.52)	*p* = 0.586
Hospital scale			*p* = 0.063
100–299 beds	1.46	(0.72–2.95)	*p* = 0.296
≥300 beds	2.01	(1.00–4.03)	*p* = 0.051
ICU, HCU	2.17	(1.28–3.70)	*p* < 0.004 **
ER	1.87	(0.89–3.93)	*p* = 0.099
Cardiology department	1.67	(1.11–2.52)	*p* = 0.015 *
Thoracic surgery, Cardiovascular surgery, Respiratory surgery department	1.75	(1.06–2.90)	*p* = 0.030 *

ECG: electrocardiogram; CI: confidence interval; ICU: intensive care unit; HCU: high care unit; ER: emergency room. Based on logistic analysis, * *p* < 0.05; ** *p* < 0.01.

## Data Availability

All deidentified participant data used in this study are available on reasonable request by contacting Sho Nishiguchi.

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
