# Peer review of "Effects of Electrocardiographic Monitoring Education on Nurses’ Confidence and Psychological Stress: An Online Cross-Sectional Survey in Japan"

_ijerph, 2022, doi:10.3390/ijerph19084742_

Round 1

Reviewer 1 Report

Both the Introduction and the conclusions can be improved. You must make them longer, because they are very short.
Likewise, citations and bibliographical references could be more current and have a greater impact.
For the rest, it is an excellent article, well structured and with an impeccable methodology and presentation and analysis of the results. Congratulations.

Author Response

Thank you for your comment. We changed our manuscript according to your suggestion.

Reviewer 2 Report

Dear Authors, your paper is quit well written. Sampling is large enough, and with a lot of queries.

Despite the initial aim and your main research query you conclude with another secondary finding ("Less than half of the study participants received pre-graduate ECG monitoring education). It would be better to put your primary conclusion first according to the aim and the title.

Generally the statistic is acceptable and you have a good discussion.

No other comments.

Author Response

We appreciate your review. Our manuscript was changed from your appropriate comments. 

Reviewer 3 Report

First of all, I would like to thank the authors for the opportunity to review this manuscript. 

The title is informative; the abstract provides a summary of the manuscript's major aspects.  

Background:

In general, the background chapter should review the current literature on the subject being investigated, as well as, capturing key studies in the field and expose the knowledge gap. There are several claims but we do not know which authors say it. For examples:

“Nurses play an important role in the management of patients’ ECG monitoring. They perform the following difficult roles: which are not easy: assessing which patients require ECG monitoring, correctly placing ECG monitoring equipment on patients, quickly de testing abnormal data, and deciding whether to intervene in clinical practice.” or “These tasks require confidence; medical care providers experience more stress due to life-threatening issues”

Also, the authors cite a study that says nurses are often the first provider to evaluate hospitalized patients presenting with abnormalities in ECG monitoring...and the authors exposes that nurses have an enormous responsibility and must make important clinical decisions, but they do not refer to any study that concludes this (line 58). Moreover, reference 2 is about using electrocardiogram monitoring to detect myocardial ischemia and maybe these conclusions couldn’t be extrapolated in general populations. In this sense I encourage the authors to be more clear when they expose conclusions from other studies.

Another example is the sentence: (Line 73) “Without ECG monitoring education, nurses cannot gain confidence, and they may 73 thus find it stressful to care for patients using continuous ECG monitoring”. Which study concludes it?

Line 64. The American Heart Association proposed practice standards for improving hospital 64 ECG monitoring in 2004. No newer standards since 2004?

Finally, in line 70 the authors talk about the contents of pre-graduate ECG monitoring education in nursing schools, and they affirm that difference between nursing schools had been detected among these training.  It would be interesting to read more about this

In summary, I recommend the authors to revise the background chapter on both the basis of more literature about the topic and more recent literature.

Material and Methods:

I suppose that the sampling of the study was an intentional one. Is that the case?

I am concerned that participants would have a reward for responding to the survey. This could not favor the response of those nurses who use the NURSE SENKA website? How did you control this point?

I think it’s important to understand how the post-graduate education was measured and it’s not clear to me. Perhaps adding the questioner would help clarify this point

Results:

I would like to congratulate the authors for the clear exposure of the results.

Discussion:

In general, under my point of view, the authors should discuss more the findings from other studies up against their own results. The main findings should then be discussed in relation to its meaning, importance and relevance (to whom or what), and in the light relevant and updated knowledge (literature). I encourage the authors to work a little more this section, providing new bibliography.

Conclusions:

The findings are clear but I miss what aspects the authors propose to promote confidence and less psychological stress among nurses through  electrocardiographic monitoring education.

Author Response

Response to Reviewer 3 Comments

Point 1: Background:

In general, the background chapter should review the current literature on the subject being investigated, as well as, capturing key studies in the field and expose the knowledge gap. There are several claims but we do not know which authors say it. For examples:

“Nurses play an important role in the management of patients’ ECG monitoring. They perform the following difficult roles: which are not easy: assessing which patients require ECG monitoring, correctly placing ECG monitoring equipment on patients, quickly de testing abnormal data, and deciding whether to intervene in clinical practice.” or “These tasks require confidence; medical care providers experience more stress due to life-threatening issues”

Response 1-1: According to comments, we added new reference of [2] and revised setences.

Also, the authors cite a study that says nurses are often the first provider to evaluate hospitalized patients presenting with abnormalities in ECG monitoring...and the authors exposes that nurses have an enormous responsibility and must make important clinical decisions, but they do not refer to any study that concludes this (line 58). Moreover, reference 2 is about using electrocardiogram monitoring to detect myocardial ischemia and maybe these conclusions couldn’t be extrapolated in general populations. In this sense I encourage the authors to be more clear when they expose conclusions from other studies.

Response 1-2: Thank you for your appropriate comment. We added reference in line 58. We changed the sentence from “However, more than half such nurses are not adequately skilled in ECG monitoring” to “In a Chinese national survey, more than half such nurses are not adequately skilled in ECG monitoring to detect myocardial ischemia”

Another example is the sentence: (Line 73) “Without ECG monitoring education, nurses cannot gain confidence, and they may 73 thus find it stressful to care for patients using continuous ECG monitoring”. Which study concludes it?

Response 1-3: According to your important comment, we converted the sentence toWe hypothesis that without ECG monitoring education, nurses cannot gain confidence, and they may thus find it stressful to care for patients using continuous ECG monitoring.”

Line 64. The American Heart Association proposed practice standards for improving hospital 64 ECG monitoring in 2004. No newer standards since 2004?

Response 1-4: Thank you for valuable comment. We convereted the citation to 2017 version. “Sandau, K.E.; Funk, M.; Auerbach, A.; et al. Update to Practice Standards for Electrocardiographic Monitoring in Hospital Settings: A Scientific Statement From the American Heart Association. Circulation 2017, 136(19), e273-e344.” 

Finally, in line 70 the authors talk about the contents of pre-graduate ECG monitoring education in nursing schools, and they affirm that difference between nursing schools had been detected among these training.  It would be interesting to read more about this

Response 1-5: According to the comment, We described sentences as“Japanese Ministry of Health, Labour and Welfare made The Bachelor of Nursing program at 255 universities in 2017, to rise level of nursing school education [7]. Based on the program, each nursing school makes an education curriculum, but ECG monitoring is not required course in the Bachelor of Nursing program. Therefore, the contents of pre-graduate ECG monitoring education differ among nursing schools.”

In summary, I recommend the authors to revise the background chapter on both the basis of more literature about the topic and more recent literature.

Response 1-6: Thank you for your important suggestions. We changed the chapter.

Point 2: Material and Methods:

I suppose that the sampling of the study was an intentional one. Is that the case?

Response 2-1: Than you for valuable comment. The study intentionally planned 2500 sample size. We added “Before initiation of the joint research study, we calculated the number of participants based on seing number of NURSE SENKA website. The study’s margin of error was plus or minus 2%.”

I am concerned that participants would have a reward for responding to the survey. This could not favor the response of those nurses who use the NURSE SENKA website? How did you control this point?

Response 2-2: As your appropriate comment, reward may occur recruitment bias. We described the point in the limitiation paragraph of Discussion part.

I think it’s important to understand how the post-graduate education was measured and it’s not clear to me. Perhaps adding the questioner would help clarify this point.

Response 2-3: According to your suggestion, we added questioner in Supplementary Material.

Point 3: Discussion:

In general, under my point of view, the authors should discuss more the findings from other studies up against their own results. The main findings should then be discussed in relation to its meaning, importance and relevance (to whom or what), and in the light relevant and updated knowledge (literature). I encourage the authors to work a little more this section, providing new bibliography.

Response 3: Thank you for your suggestions, we added new reference and described meaning of main findings in a Discussion part.

Point 4: Conclusions:

The findings are clear but I miss what aspects the authors propose to promote confidence and less psychological stress among nurses through electrocardiographic monitoring education.

Response 4: We appreciate valuable comment. According to your suggestion, we converted conclusions to “Post-graduate ECG monitoring education experience was associated with nurses’ improved confidence regarding continuous ECG monitoring. However, the experience of post-graduate education had no effect on nurses’ psychological stress regarding continuous ECG monitoring. Less than half of the study participants received pre-graduate ECG monitoring education. Pre-graduate education can provide more chance to learn ECG monitoring for nurses and post-graduate education may contribute giving nurses’ confidence.”

Round 2

Reviewer 3 Report

I would like to congratulate all the authors for the modifications made in this paper. Under my point of view the article has been greatly improved.